# Cross Attention Network for Few-shot Classification

**Ruibing Hou**[1,2], **Hong Chang**[1,2], **Bingpeng Ma**[2], **Shiguang Shan**[1,2,3], **Xilin Chen**[1,2]

[1]Key Laboratory of Intelligent Information Processing of Chinese Academy of Sciences (CAS),
Institute of Computing Technology, CAS, China
[2]University of Chinese Academy of Sciences, China
[3]CAS Center for Excellence in Brain Science and Intelligence Technology, China
ruibing.hou@vipl.ict.ac.cn, {changhong, sgshan,xlchen}@ict.ac.cn, bpma@ucas.ac.cn

## Abstract

Few-shot classification aims to recognize unlabeled samples from *unseen classes* given only *few labeled* samples. The unseen classes and low-data problem make few-shot classification very challenging. Many existing approaches extracted features from labeled and unlabeled samples independently, as a result, the features are not discriminative enough. In this work, we propose a novel *Cross Attention Network* to address the challenging problems in few-shot classification. Firstly, *Cross Attention Module* is introduced to deal with the problem of unseen classes. The module generates cross attention maps for each pair of class feature and query sample feature so as to highlight the target object regions, making the extracted feature more discriminative. Secondly, a transductive inference algorithm is proposed to alleviate the low-data problem, which iteratively utilizes the unlabeled query set to augment the support set, thereby making the class features more representative. Extensive experiments on two benchmarks show our method is a simple, effective and computationally efficient framework and outperforms the state-of-the-arts.

## 1 Introduction

Few-shot classification aims at classifying unlabeled samples (query set) into *unseen classes* given very *few labeled* samples (support set). Compared to traditional classification, few-shot classification has two main challenges: One is unseen classes, *i.e.*, the non-overlap between training and test classes; The other is the low-data problem, *i.e.*, very few labeled samples for the test unseen classes.

Solving few-shot classification problem requires the model trained with seen classes to generalize well to unseen classes with only few labeled samples. A straightforward approach is fine-tuning a pre-trained model using the few labeled samples from the unseen classes. However, it may cause severe overfitting. Regularization and data augmentation can alleviate but cannot fully solve the overfitting problem. Recently, meta-learning paradigm [38, 39, 22] is widely used for few-shot learning. In meta-learning, the transferable meta-knowledge, which can be an optimization strategy [31, 1], a good initial condition [7, 16, 24], or a metric space [35, 40, 37], is extracted from a set of training tasks and generalizes to new test tasks. The tasks in the training phase usually mimic the settings in the test phase to reduce the gap between training and test settings and enhance the generalization ability of the model.

While promising, few of them pay enough attention to the discriminability of the extracted features. They generally extract features from the support classes and unlabeled query samples independently, as a result, the features are not discriminative enough. For one thing, **the test images in the support/query set are from *unseen classes*, thus their features can hardly attend to the target objects**. To be specific, for a test image containing multiple objects, the extracted feature may attend to the objects from seen classes which have large number of labeled samples in the training set, while ignore the target object from unseen class. As illustrated in Fig. 1 (c) and (d), for the two

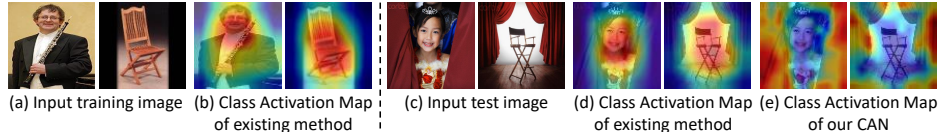

| (a) Input training image | (b) Class Activation Map of existing method | (c) Input test image | (d) Class Activation Map of existing method | (e) Class Activation Map of our CAN |

Figure 1. An example of the class activation maps [48] of training and test images of existing method [35] and our method CAN. Warmer color with higher value.

images from the test class *curtain*, the extracted features only capture the information of the objects that are related to the training classes, such as person or chair in Fig. 1 (a) and (b). For another, **the low-data problem makes the feature of each test class not representative for the true class distribution, as it is obtained from very *few labeled* support samples**. In a word, the independent feature representations may fail in few-shot classification.

In this work, we propose a novel *Cross Attention Network* (CAN) to enhance the feature discriminability for few-shot classification. Firstly, *Cross Attention Module* (CAM) is introduced to deal with the *unseen class* problem. The cross attention idea is inspired by the human few-shot learning behavior. To recognize a sample from unseen class given a few labeled samples, human tends to firstly locate the most relevant regions in the pair of labeled and unlabeled samples. Similarly, given a class feature map and a query sample feature map, CAM generates a cross attention map for each feature to highlight the target object. Correlation estimation and meta fusion are adopted to achieve this purpose. In this way, the target object in the test samples can get attention and the features weighted by the cross attention maps are more discriminative. As shown in Fig. 1 (e), the extracted features with CAM can roughly localize the regions of target object curtain. Secondly, we introduce a transductive inference algorithm that utilizes the entire unlabeled query set to alleviate the low-data problem. The proposed algorithm iteratively predicts the labels for the query samples, and selects pseudo-labeled query samples to augment the support set. With more support samples per class, the obtained class features can be more representative, thus alleviating the low-data problem.

Experiments are conducted on multiple benchmark datasets to compare the proposed CAN with existing few-shot meta-learning approaches. Our method achieves new state-of-the art results on all dataset, which demonstrates the effectiveness of CAN.

## 2 Related Work

**Few-Shot Classification.** On the basis of the availability of the entire unlabeled query set, few-shot classification can be divided into two categories: *inductive* and *transductive* few-shot classification. In this work, we mainly explore the few-shot approaches based on meta-learning.

Inductive Few-shot Learning has been a well studied area in recent years. One promising way is the meta-learning [38, 39, 22] paradigm. It usually trains a meta-learner from a set of tasks, which extracts meta-knowledge to transfer into new tasks with scarce data. Meta learning approaches for few-shot classification can be roughly categorized into three groups. *Optimization-based methods* designed the meta-learner as an optimizer that learned to update model parameters [1, 31, 18]. Further, the works [7, 33, 36] learned a good initialization so that the learner could rapidly adapt to novel tasks within a few optimization steps. *Parameter-generating based methods* [2, 20, 21, 3] usually designed the meta-learner as a parameter predicting network. *Metric-learning based methods* [40, 35, 37, 26, 5] learned a common feature space where categories can distinguish with each other based on a distance metric. For example, Matching Network [40] produced a weighted nearest neighbor classifier. Prototypical Network [35] performed nearest neighbor classification with learned class features (prototypes). The works [37, 26, 15] improved the prototypical network with a learnable similarity metric [37], a task adaptive metric [26], or a image-to-class local metric [5].

Our proposed framework belongs to *metric-learning based method*. Different from existing metric-learning based methods which extracted the support and query sample features independently, our method exploits the semantic relevance between support and query features to highlight the target object. Although the *parameter-generating based methods* also consider the relationship between support and query samples, these approaches require an additional complex parameter prediction network. With less overload, our approach outperforms these methods by a large margin.

**Transductive Algorithm.** Transductive few-shot classification is firstly introduced in [17], which constructed a graph on the support set and the entire query set, and propagated labels within the

graph. However, the method required a specific architecture, making it less universal. Inspired by the self-training strategy in semi-supervised learning [6, 25, 42, 32], we propose a simper and more general transductive few-shot algorithm, which explicitly augments the labeled support set with unlabeled query samples to achieve more representative class features. Moreover, the proposed transductive algorithm can be directly applied to the existing models, *e.g.*, prototypical network [35], matching network [40], and relation network [37].

**Attention Model.** Attention mechanisms aim to highlight important local regions to extract more discriminative features. It has achieved great success in computer vision applications, such as image classification [12, 41, 27], person re-identification [10, 47, 11], image caption [29, 44, 4] and visual question answering [43, 45, 46, 30]. In image classification, SENet [12] proposed a channel attention block to boost the representational power of a network. Woo *et al.* [41, 27] further integrated the channel and spatial attention modules to a block. In image caption, the attention blocks [44, 4] usually used the last *generated words* to search for related regions in the image to generate the next word. And in visual question answering, the attention blocks [43, 45, 46, 30] used the *questions* to localize the related regions in the image to answer. Specifically, [30] uses the questions to generate a convolutional kernel which is used to convolve with the image feature. On the contrary, our method uses a meta-learner to generate a kernel which is used to fuse the relations to get the final attention map. For few-shot image classification, in this paper, we design a meta-learner to compute the cross attention between support (or class) and query feature maps, which helps to locate the important regions of the target object and enhance the feature discriminability.

## 3   Cross Attention Module

**Problem Define.** Few-shot classification usually involves a training set, a support set and a query set. The training set contains a large number of classes and labeled samples. The support set of few labeled samples and the query set of unlabeled samples share the same label space, which is disjoint with that of the training set. Few-shot classification aims to classify the unlabeled query samples given the training set and support set. If the support set consists of $C$ classes and $K$ labeled samples per class, the target few-shot problem is called $C$-way $K$-shot.

Following [40, 35, 34, 19, 9, 14, 7], we adopt the episode training mechanism, which has been demonstrated as an effective approach for few-shot learning. The episodes used in training simulate the settings in test. Each episode is formed by randomly sampling $C$ classes and $K$ labeled samples per class as the support set $\mathcal{S} = \{(x_a^s, y_a^s)\}_{a=1}^{n_s}$ ($n_s = C \times K$), and a fraction of the rest samples from the $C$ classes as the query set $\mathcal{Q} = \{(x_b^q, y_b^q)\}_{b=1}^{n_q}$. And we denote $\mathcal{S}^k$ as the support subset of the $k^{th}$ class. How to represent each support class $\mathcal{S}^k$ and query sample $x_b^q$ and measure the similarity between them is a key issue for few-shot classification.

**CAM Overview.** In this work, we resort to *metric-learning* to obtain proper feature representations for each pair of support class and query sample. Different from existing methods which extract the class and query features independently, we propose *Cross Attention Module* (CAM) which can model the semantic relevance between the class feature and query feature, thus draw attention to the target objects and benefit the subsequent matching.

CAM is illustrated in Fig. 2. The class feature map $P^k \in \mathbb{R}^{c \times h \times w}$ is extracted from the support samples in $\mathcal{S}^k$ ($k \in \{1, 2, \ldots, C\}$) and the query feature map $Q^b \in \mathbb{R}^{c \times h \times w}$ is extracted from the query sample $x_b^q$ ($b \in \{1, 2, \ldots, n_q\}$), where $c$, $h$ and $w$ denote the number of channel, height and width of the feature maps respectively. CAM generates cross attention map $A^p$ ($A^q$) for $P^k$ ($Q^b$), which is then used to weight the feature map to achieve more discriminative feature representation $\bar{P}_b^k$ ($\bar{Q}_k^b$). For simplicity, we omit the superscripts and subscripts, and denote the input class and query feature maps as $P$ and $Q$, and the output class and query feature maps as $\bar{P}$ and $\bar{Q}$, respectively.

**Correlation Layer.** As shown in Fig. 2, we first design a *correlation layer* to calculate a correlation map between $P$ and $Q$, which is then used to guide the generation of the cross attention maps. To this end, we first reshape $P$ and $Q$ to $\mathbb{R}^{c \times m}$, *i.e.*, $P = [p_1, p_2, \ldots, p_m]$ and $Q = [q_1, q_2, \ldots, q_m]$, where $m$ ($m = h \times w$) is the number of spatial positions on each feature map. $p_i, q_i \in \mathbb{R}^c$ are the feature vectors at the $i^{th}$ spatial position in $P$ and $Q$ respectively. The *correlation layer* computes the semantic relevance between $\{p_i\}_{i=1}^m$ and $\{q_i\}_{i=1}^m$ with cosine distance to get the correlation map

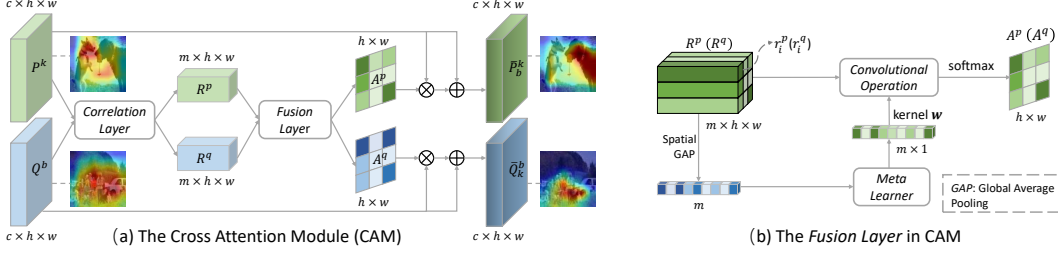

Figure 2. (a) Cross Attention Module (CAM). (b) the *Fusion Layer* in CAM. In the figure, $R^p$ ($R^q$) $\in \mathbb{R}^{m \times m}$ is reshaped to $\mathbb{R}^{m \times h \times w}$ for a better visualization. As seen, CAM can generate the feature maps that attend to the regions of target object (*coated retriever* in the figure).

$R \in \mathbb{R}^{m \times m}$ as:

$$R_{ij} = \left( \frac{p_i}{||p_i||_2} \right)^T \left( \frac{q_j}{||q_j||_2} \right), \quad i, j = 1, \dots, m. \tag{1}$$

Furthermore, we define two correlation maps based on $R$: the class correlation map $R^p \doteq R^T = [r_1^p, r_2^p, \dots, r_m^p]$ and the query correlation map $R^q \doteq R = [r_1^q, r_2^q, \dots, r_m^q]$, where $r_i^p \in \mathbb{R}^m$ denotes the relevance between the local class feature vector $p_i$ and all query feature vectors $\{q_i\}_{i=1}^m$, and $r_i^q \in \mathbb{R}^m$ is the relevance between local query feature vector $q_i$ and all class feature vectors $\{p_i\}_{i=1}^m$. In this way, $R^p$ and $R^q$ characterize the local correlations between the class and query feature maps.

**Meta Fusion Layer.** A *meta fusion layer* is then used to generate the class and query attention maps, respectively, based on the corresponding correlation maps. We take the class attention map as an example. As shown in Fig. 2 (b), the fusion layer takes the class correlation map $R^p$ as input, and applies convolutional operation with a $m \times 1$ kernel, $w \in \mathbb{R}^{m \times 1}$, to fuse each local correlation vector $\{r_i^p\}$ of $R^p$ into an attention scalar. A softmax function is then used to normalize the attention scalar to obtain the class attention at the $i^{th}$ position:

$$A_i^p = \frac{\exp\left( (w^T r_i^p)/\tau \right)}{\sum_{j=1}^{h \times w} \exp\left( (w^T r_j^p)/\tau \right)}, \tag{2}$$

where $\tau$ is the temperature hyperparameter. Lower temperature leads to lower entropy, making the distribution concentrate on a few high confidence positions. The *class attention map* is then obtained by reshaping $A^p$ to matrix in $\mathbb{R}^{h \times w}$. Note that the kernel $w$ plays a crucial role in the fusion. It aggregates the correlations between the local class feature $p_i$ and all local query features $\{q_j\}_{j=1}^m$ as the attention scalar at the $i^{th}$ position. More importantly, the weighted aggregation should draw attention to the target object, instead of simply highlighting the visually similar regions across support class and query sample.

Based on above analysis, we design a *meta-learner* to adaptively generate the kernel based on the correlation between the class and the query features. To this end, we apply global average pooling (*GAP*) operation (*i.e.*, row-wise averaging) to $R^p$ to obtain an averaged query correlation vector, which is then fed into the meta-learner to generate the kernel $w \in \mathbb{R}^m$:

$$w = W_2(\sigma(W_1(GAP(R^p)), \tag{3}$$

where $W_1 \in \mathbb{R}^{\frac{m}{r} \times m}$ and $W_2 \in \mathbb{R}^{m \times \frac{m}{r}}$ are the parameters of the meta-learner, and $r$ is the reduction ratio. $\sigma$ refers to the ReLU function [23]. The nonlinearity in the meta-learning model allows a flexible transformation. For each pair of class and query features, the meta-learner is expected to generate a kernel $w$ which can draw cross attention to the target object. This is achieved in meta training by minimizing the classification errors on the query samples.

In a similar way, we can get the *query attention map* $A^q \in \mathbb{R}^{h \times w}$. At last, we use a residual attention mechanism, where the initial feature maps $P$ and $Q$ are elementwisely weighted by $1 + A^p$ and $1 + A^q$, to form more discriminative feature maps $\bar{P} \in \mathbb{R}^{c \times h \times w}$ and $\bar{Q} \in \mathbb{R}^{c \times h \times w}$, respectively.

**Complexity Analysis.** The time and space cost of CAM is mainly on *correlation layer*. The time complexity of CAM is $O(h^2 w^2 c)$ and the space complexity is $O(hwc)$, which both varies with the size of input feature map. So we insert CAM after the last convolutional layer to avoid excessive cost.

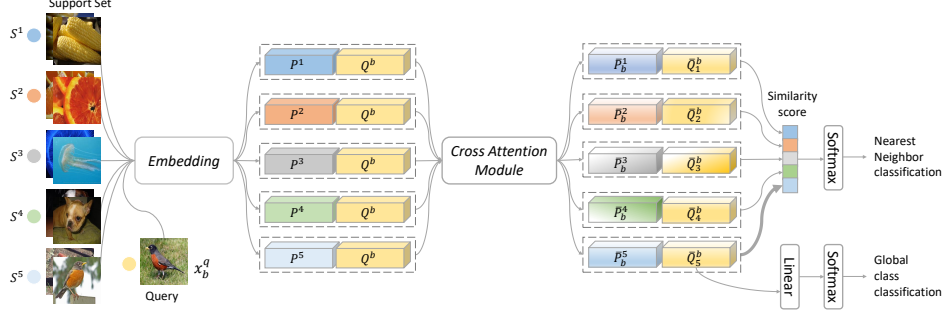

Figure 3. The framework of the proposed CAN approach.

## 4 Cross Attention Network

The overall *Cross Attention Network* (CAN) is illustrated in Fig. 3, which consists of three modules: an *embedding* module, a *cross attention module* and a *classification module*. The embedding module $E$ consists of several cascaded convolutional layers, which maps an input image $x$ into a feature map $E(x) \in \mathbb{R}^{c \times h \times w}$. Following prototypical network [35], we define the class feature as the the mean of its support set in the embedding space. As shown in Fig. 3, the embedding module $E$ takes the support set $\mathcal{S}$ and a query sample $x_b^q$ as inputs, and produces the class feature map $P^k = \frac{1}{|\mathcal{S}^k|} \sum_{x_a^s \in \mathcal{S}^k} E(x_a^s)$ and a query feature map $Q^b = E(x_b^q)$. Each pair of feature maps ($P^k$ and $Q^b$) are then fed through the cross attention module, which highlights the relevant regions and outputs more discriminative feature pairs ($\bar{P}_b^k$ and $\bar{Q}_k^b$) for classification.

**Model Training via Optimization.** CAN is trained via minimizing the classification loss on the query samples of training set. The classification module consists of a nearest neighbor and a global classifier. The nearest neighbor classifier classifies the query samples into $C$ support classes based on pre-defined similarity measure. To obtain precise attention maps, we constrain each position in the query feature maps to be correctly classified. Specifically, for each local query feature $q_i^b$ at $i^{th}$ position, the nearest neighbor classifier produces a softmax-like label distribution over $C$ support classes. The probability of predicting $q_i^b$ as $k^{th}$ class is:

$$p(y = k|q_i^b) = \frac{\exp\left(-d\left((\bar{Q}_k^b)_i, GAP(\bar{P}_b^k)\right)\right)}{\sum_{j=1}^{C} \exp\left(-d\left((\bar{Q}_j^b)_i, GAP(\bar{P}_b^j)\right)\right)}, \tag{4}$$

where $(\bar{Q}_k^b)_i$ denotes the feature vector in the $i^{th}$ spatial position of $\bar{Q}_k^b$, and *GAP* is the global average pooling operation to get the mean class feature. Note that $\bar{Q}_k^b$ and $\bar{Q}_j^b$ represent the query sample $x_b^q$ from somewhat different views as they are correlated with different support classes. In Eq. 4, the cosine distance $d$ is calculated in the feature space generated by CAM. The nearest neighbor classification loss is then defined as the negative log-probability according to the true class label $y_b^q \in \{1, 2, \dots, C\}$:

$$L_1 = -\sum_{b=1}^{n_q} \sum_{i=1}^{m} \log p(y = y_b^q|q_i^b). \tag{5}$$

The global classifier uses a fully connected layer followed by *softmax* to classify each query sample among all available training classes. Suppose there are overall $l$ classes in the training set. The classification probability vector $z_i^b \in \mathbb{R}^l$ for each local query feature $q_i^b$ is computed as $z_i^b = softmax(W_c(\bar{Q}_{y_b^q}^b)_i)$. The global classification loss is then expressed as:

$$L_2 = -\sum_{b=1}^{n_q} \sum_{i=1}^{m} \log\left((z_i^b)_{l_b^q}\right), \tag{6}$$

where $W_c \in \mathbb{R}^{l \times c}$ is the weight of the fully connected layer and $l_b^q \in \{1, 2, \dots, l\}$ is the true global class of $x_b^q$. Finally, the overall classification loss is defined as $L = \lambda L_1 + L_2$, where $\lambda$ is the weight to balance the effects of different losses. The network can be trained end-to-end by optimizing $L$ with gradient descent algorithm.

**Inductive Inference.** In inductive inference phase, the embedding module is directly used for a novel task to extract the class and query feature maps. Then each pair of class and query feature maps are fed into CAM to get the attention weighted features. The global averaging pooling is then

performed to the outputs of CAM to get the mean class and query features. Finally, the label $\hat{y}_b^q$ for a query sample $x_b^q$ is predicted by finding the nearest mean class feature under cosine distance metric:

$$\hat{y}_b^q = \arg\min_k d\left(GAP(\bar{Q}_k^b), GAP(\bar{P}_b^k)\right) \tag{7}$$

**Transductive Inference.** In few-shot classification task, each class has very few labeled samples, so the class feature can hardly represent the true class distribution. In order to alleviate the problem, we propose a simple and effective transductive inference algorithm which utilizes the unlabeled query samples to enrich the class feature.

Specifically, we firstly utilize the initial class feature map $P^k$ to predict the labels $\{\hat{y}_b^q\}_{b=1}^{n_q}$ of the unlabeled query samples $\{x_b^q\}_{b=1}^{n_q}$ using Eq. 7. Then, we define a label confidence criterion using the cosine distance between the query sample $x_b^q$ and its nearest class neighbor: $c_b^q = \min_k d(GAP(\bar{Q}_k^b), GAP(\bar{P}_b^k))$. The lower the value $c_b^q$, the higher the confidence of the predicted label $\{\hat{y}_b^q\}$. Based on this criterion, we can obtain a candidate set $\mathcal{D} = \{(x_b^q, \hat{y}_b^q)|s_b = 1, x_b^q \in \mathcal{Q}\}$, where $s_b \in \{0,1\}$ denotes the selection indicator for the query sample $x_b^q$. The selection indicator $s \in \{0,1\}^{n_q}$ is determined by the top $t$ confident query samples: $s = \arg\min_{||s||_0=t} \sum_{b=1}^{n_q} s_b c_b^q$. Finally, the candidate set $\mathcal{D}$ along with the support set $\mathcal{S}$ is used to generate a more representative class feature map $(P^k)^*$:

$$(P^k)^* = \frac{1}{|\mathcal{S}^k| + |\mathcal{D}^k|}\left(\sum_{x_a^s \in \mathcal{S}^k} E(x_a^s) + \sum_{x_b^q \in \mathcal{D}^k} E(x_b^q)\right). \tag{8}$$

Here $\mathcal{D}^k = \{(x_b^q, \hat{y}_b^q)|x_b^q \in \mathcal{D}, \hat{y}_b^q = k\}$. $(P^k)^*$ is then used to re-estimate the pseudo label for each query sample. We repeat above process for a certain number of iterations. And the number of selected samples in the candidates set $\mathcal{D}$ is gradually increased with a fixed ratio in each iteration. In this way, we can progressively enrich the class features to be more representative and robust.

## 5 Experiments

### 5.1 Experiment Setup

**Datasets.** We use **miniImageNet** [40] which is a subset of ILSVRC-12 [13]. It contains 100 classes with 600 images per class. We use the standard split following [31, 37, 26, 15, 33]: 64 classes for training, 16 for validation and 20 for testing. We also use **tieredImageNet** dataset [32], a much larger subset of ILSVRC-12 [13]. It contains 34 categories and 608 classes in total. These are divided into 20 categories (351 classes) for training, 6 categories (97 classes) for validation, and 8 categories (160 classes) for testing, as in [32, 7, 35, 37].

**Experimental setting.** We experiment our approach on 5-way 1-shot and 5-way 5-shot settings. For a $C$-way $K$-shot setting, the episode is formed with $C$ classes and each class includes $K$ support samples, and 6 and 15 query samples are used for training and inference respectively. When inference, 2000 episodes are randomly sampled from the test set. We report the *average accuracy* and the corresponding $95\%$ *confidence interval* over the 2000 episodes.

**Implementation details.**[1] Pytorch [28] is used to implement all our experiments on one NVIDIA 1080Ti GPU. Following [26, 17, 36, 21], we use ResNet-12 network as our embedding module. The input images size is $84 \times 84$. During training, we adopt horizontal flip, random crop and random erasing [49] as data augmentation. SGD is used as the optimizer. Each mini-batch contains 8 episodes. The model is trained for 80 epochs, with each epoch consisting of $1,200$ episodes. For miniImageNet, the initial learning rate is 0.1 and decreased to 0.006 and 0.0012 at 60 and 70 epochs, respectively. For tieredImageNet, the initial learning rate is set to 0.1 with a decay factor 0.1 at every 20 epochs. The temperature hyperparameter ($\tau$ in Eq. 3) is set to 0.025, the reduction ratio in the meta-learner is set to 6, and the weight hyperparameter ($\lambda$) in the overall loss function is set to 0.5. For the transductive algorithm, the selected number of query samples in the first iteration ($t$) is set to 35, and the number of iterations and enlarging factor of candidate set are both set to 2. All hyperparameters are cross-validated in the validation sets and fixed afterwards in all experiments.

Table 1. Comparison to state-of-the-arts with 95% confidence intervals on 5-way classification on miniImageNet and tieredImageNet datasets. IT: Inference Time per query data in a 5-way 1-shot task on one NVIDIA 1080Ti GPU. **CAN+T** denotes CAN with transductive inference. The methods are separated into four groups: optimization-based (**O**), parameter-generating (**P**), metric-learning (**M**) and transductive methods (**T**).

| | model | Embedding | IT(s) | miniImageNet | | tieredImageNet | |
| --- | --- | --- | --- | --- | --- | --- | --- |
| | | | | 1-shot | 5-shot | 1-shot | 5-shot |
| **O** | MAML [7] | ConvNet | 0.103 | $48.70 \pm 0.84$ | $55.31 \pm 0.73$ | $51.67 \pm 1.81$ | $70.30 \pm 1.75$ |
| | MTL [36] | ResNet-12 | 2.020 | $61.20 \pm 1.80$ | $75.50 \pm 0.80$ | - | - |
| | LEO [33] | WRN-28 | - | $61.76 \pm 0.08$ | $77.59 \pm 0.12$ | $66.33 \pm 0.05$ | $81.44 \pm 0.09$ |
| | MetaOpt [14] | ResNet-12 | 0.096 | $62.64 \pm 0.62$ | $78.63 \pm 0.46$ | $65.99 \pm 0.72$ | $81.56 \pm 0.53$ |
| **P** | MetaNet [20] | ConvNet | - | $49.21 \pm 0.96$ | - | - | - |
| | MM-Net [3] | ConvNet | - | $53.37 \pm 0.48$ | $66.97 \pm 0.35$ | - | - |
| | adaNet [21] | ResNet-12 | 1.371 | $56.88 \pm 0.62$ | $71.94 \pm 0.57$ | - | - |
| **M** | MN [40] | ConvNet | 0.021 | $43.44 \pm 0.77$ | $60.60 \pm 0.71$ | - | - |
| | PN [35] | ConvNet | **0.018** | $49.42 \pm 0.78$ | $68.20 \pm 0.66$ | $53.31 \pm 0.89$ | $72.69 \pm 0.74$ |
| | RN [37] | ConvNet | 0.033 | $50.44 \pm 0.82$ | $65.32 \pm 0.70$ | $54.48 \pm 0.93$ | $71.32 \pm 0.78$ |
| | DN4 [15] | ConvNet | 0.049 | $51.24 \pm 0.74$ | $71.02 \pm 0.64$ | - | - |
| | TADAM [26] | ResNet-12 | 0.079 | $58.50 \pm 0.30$ | $76.70 \pm 0.30$ | - | - |
| | **Our CAN** | ResNet-12 | 0.044 | $\mathbf{63.85 \pm 0.48}$ | $\mathbf{79.44 \pm 0.34}$ | $\mathbf{69.89 \pm 0.51}$ | $\mathbf{84.23 \pm 0.37}$ |
| **T** | TPN [17] | ResNet-12 | - | 59.46 | 75.65 | - | - |
| | **Our CAN+T** | ResNet-12 | - | $\mathbf{67.19 \pm 0.55}$ | $\mathbf{80.64 \pm 0.35}$ | $\mathbf{73.21 \pm 0.58}$ | $\mathbf{84.93 \pm 0.38}$ |

## 5.2 Comparison with State-of-the-arts

Tab. 1 compares our method with existing few-shot methods on miniImageNet and tieredImageNet. The comparative methods are categorized into four groups, *i.e.*, optimization-based methods (**O**), parameter-generating methods (**P**), metric-learning methods (**M**), and transductive methods (**T**). Our method outperforms the *optimization-based methods* [7, 18, 33, 36]. It is noted that the optimization-based methods need fine-tuning on the target task, making the classification time consuming. On the contrary, our method requires no model updating solves the tasks in an feed-forward manner, which is much faster and simpler than above methods and has better results.

Our method performs better than the *parameter-generating methods* [20, 21, 3], with an improvement up to $7\%$. These approaches generate the parameters of the feature extractor based on the support set and extract the query features adaptively. However, these methods suffer from the high dimensionality of the parameter space. Instead, our method uses a cross attention module to adaptively extract the support and query features, which is computationally lightweight and achieves a better performance. Our method belongs to the *metric-learning methods*. Existing metric-based methods [40, 35, 37, 26, 15] extract features of support and query samples independently, making the features attend to non-target objects. Instead, our CAN highlights the target object regions and gets more discriminative features. Compared to TADAM [26], CAN with almost the same number of parameters achieves $5\%$ higher performance on 1-shot, which demonstrates the superiority of our cross attention module.

In the *transductive setting*, CAN with transduction (**CAN+T**) outperforms the prior work TPN [17] by a large margin, up to $8\%$ and $5\%$ improvements on 1-shot and 5-shot respectively. TPN uses a graph network to propagate the labels of the support set to the query set. In contrast, our algorithm selects the top confident query samples to augment the support set, which can explicitly alleviate the low-data problem. In addition, our transductive algorithm can be easily applied to other few-shot learning models, *e.g.*, matching network [40], prototypical network [35] and relation network [37].

**Time complexity comparison.** Tab. 1 further compares the time cost of our method to others. Some methods [40, 35, 37, 15, 7] use a 4-layer ConvNet as the backbone thus take relatively lower time cost. Even though, our CAN is still comparable even superior to these methods in term of time cost, with a performance improvement up to $10\%$. The others use the same backbone as CAN, but require following up modules such as model update per task [36, 14], gradient-based parameter generation [21], or expensive condition generation [26], which all incur more time overhead than CAM. Overall, Tab. 1 shows that CAN outperforms other methods without excessive overhead.

## 5.3 Ablation Study

In this subsection, we empirically show the effectiveness of each component of CAN. We firstly introduce two baselines to be used for comparison. In **R12-proto** [35], the features from embedding

Table 2. Ablation study on miniImageNet and complexity comparisons. PN: Parameter Number; GFLOPs: the number of floating-point operations; CIT: CPU Inference Time of a task with 15 query samples per class.

| Description | PN | 5-way 1-shot | | | 5-way 5-shot | | |
|---|---|---|---|---|---|---|---|
| | | GFLOPs | CIT | accuracy | GFLOPs | CIT | accuracy |
| R12-proto | 8.04M | 101.550 | 0.96s | 55.46 | 126.938 | 1.25s | 69.00 |
| R12-proto-ac | 8.04M | 101.550 | 0.97s | 61.30 | 126.938 | 1.26s | 76.70 |
| CAN-NoML-1 | 8.04M | 101.812 | 1.01s | 63.55 | 127.201 | 1.29s | 78.88 |
| CAN-NoML-2 | 8.04M | 101.812 | 1.01s | 63.38 | 127.201 | 1.30s | 79.08 |
| **CAN** | 8.04M | 101.813 | 1.02s | **63.85** | 127.203 | 1.31s | **79.44** |
| **CAN+T** | 8.04M | 101.930 | 1.11s | **67.19** | 127.320 | 1.43s | **80.64** |

Table 3. The proposed transductive algorithm for other few-shot learning models. * indicates our re-implemented results using the code provided by LwoF [8]. () indicates the results reported in the paper.

| Models | Inductive | | Transductive | |
|---|---|---|---|---|
| | 1-shot | 5-shot | 1-shot | 5-shot |
| Matching Network [40] | 53.52* (43.77) | 66.20* (60.60) | 56.31 | 69.80 |
| Prototypical Network [35] | 53.68* (49.42) | 70.44* (68.20) | 55.15 | 71.12 |
| Relation Network [37] | 50.65* (50.44) | 64.18* (65.32) | 52.40 | 65.36 |

module are directly fed to the nearest neighbor classifier and the model is trained with nearest neighbor classification loss. In **R12-proto-ac**, the only difference from R12-proto is that R12-proto-ac has an additional logit head for global classification (the normal 64-way classification in miniImageNet case) and the model is trained with the joint of global and nearest neighbor classification loss.

**Influence of global classification.** The comparison results are shown in Tab. 2. By comparing R12-proto-ac to R12-proto, we can find large improvements on both 1-shot (5.8%) and 5-shot (7.7%). We further try another meta-learner matching network (MN) [40][2], and the proposed joint learning schema improves MN from 55.29% to 59.14% on 1-shot setting and 67.74% to 73.81% on 5-shot setting. The consistent improvements demonstrate the effectiveness of the joint leaning schema. We argue that the global classification loss provides regularization on the embedding module and forces it to perform well on two decoupled tasks, nearest neighbor classification and global classification.

**Influence of cross attention module.** By comparing our CAN to R12-pro-ac, we observe consistent improvements on both 1-shot and 5-shot scenarios. The reason is that when using the cross attention module, our model is able to highlight the relevant regions and extract more discriminative feature. The performance gap also provides evidence that (1) conventionally independently extracted features tend to focus on non-target region and produce inaccurate similarities. (2) cross attention module can help to highlight target regions and reduce such inaccuracy with small overhead.

**Influence of meta-learner in CAM.** To verify the effectiveness of the meta-learner in CAM, we develop two variants of CAM without meta-learner. Specifically, one variant named **CAN-NoML-1** sets the kernel $w$ (shown in Fig. 2 (b)) to be a fixed mean kernel, *i.e.*, performing global average pooling on the correlation map $R$ to get the attention maps $A$. The other variant, **CAM-NoML-2**, sets the kernel $w$ to a vanilla learnable convolutional kernel that remains the same for all input samples. As shown in Tab. 2, both variants outperform R12-proto-ac consistently, which further demonstrates the effectiveness of the proposed cross attention mechanism. The improvements of CAN-NoML-1 shows the mean of correlators can roughly estimate the relevant semantic information, which furthers verifies the reasonability of our designed meta-leaner. As seen, CAN outperforms both variants. The improvement can be attributed to the meta-learning schema which learns to adaptively generate the kernel $w$ according to the input pair of feature maps.

**Influence of transductive inference algorithm.** As shown in Tab. 2, CAN+T greatly improves CAN especially in 1-shot where the low-data problem is more serious. To further verity its effectiveness, we apply it to other few-shot models, *i.e.*, matching network [40], prototypical network [35] and relation network [37]. We re-implement these models using the code provided by [8] to ensure a fair comparison. As shown in Tab. 3, our algorithm consistently improves the performance of these models, which demonstrates its generalization ability. Nevertheless, the improvements to these models are inferior to CAN. We argue that CAN can predict more precise pseudo labels for query samples and augment the support set more effectively, thus leading to better performance.

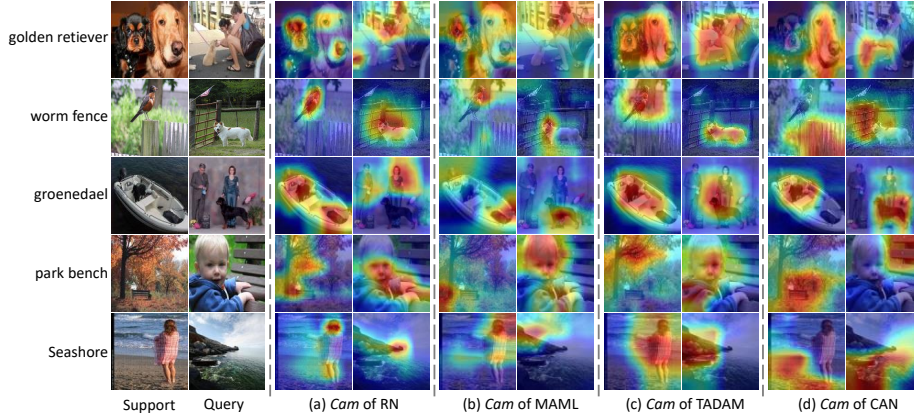

| | | (a) *Cam* of RN | (b) *Cam* of MAML | (c) *Cam* of TADAM | (d) *Cam* of CAN |

Figure 4. Class activation mapping (*Cam*) visualization on a 5-way 1-shot task with 1 query sample per class.

**Complexity comparisons.** To illustrate the cost of CAN, we report the number of parameters (PN), the number of floating-point operations (GFLOPs) and the average CPU inference time (CIT) for a 5-way 1-shot and 5-way 5-shot task with 15 query samples per class. As shown in Tab. 2, CAN introduces negligible parameters (the parameters $W_1$ and $W_2$ of the meta-learner in CAM) and small computational overhead. For example, CAN requires 101.81 GFLOPs for 5-way 1-shot, corresponding to only 0.25% relative increase over original R12-proto-ac. Notably, the correlation map in CAM can be worked out by one matrix multiplication, which occupies less time in GPU libraries. The transductive inference algorithm also introduces small computational overhead (0.37% on 1-shot and 0.31% on 5-shot) since it directly utilizes the extracted embedding features to regenerate the class feature and only passes the lightweight CAM again.

## 5.4 Visualization Analysis

To qualitatively evaluate the proposed cross attention mechanism, we compare the class activation maps [48] visualization results of CAN to other meta-learners, RN [37], MAML [7] and TADAM [26]. As shown in Fig. 4 (a), the features of RN usually contain non-target objects since it lacks an explicit mechanism for feature adaptation. MAML performs gradient-based adaptation, which makes the model merely learn some conspicuous discriminative features in the support images without deeping into the intrinsic characteristic of the target objects. As shown in Fig. 4 (b), MAML attends to *ship* for the *groenendael* support image to better distinguish it from the *golden retriever* category, resulting in a confusing location and misclassification of the *groenendael* category. TADAM performs task-dependent adaptation and applies the **same** adaptive parameters to all query images of a task, thus it is difficult to locate different target objects for different categories. As shown in Fig. 4 (c), TADAM mistakenly attends to the *dog* for *worm fence* query image. In contrast, CAN processes the query samples with **different** adaptive parameters, which allows it to focus on the different target objects for different categories shown in Fig. 4 (d).

## 6 Conclusion

In this paper, we proposed a cross attention network for few-shot classification. Firstly, a cross attention module is designed to model the semantic relevance between class and query features. It can adaptively localize the relevant regions and generate more discriminative features. Secondly, we propose a transductive inference algorithm to alleviate the low-data problem. It utilizes the unlabeled query samples to enrich the class features to be more representative. Extensive experiments show that our method is far simpler and more efficient than recent few-shot meta-learning approaches, and produces state-of-the-art results.

**Acknowledgement** This work is partially supported by National Key R&D Program of China (No.2017YFA0700800), Natural Science Foundation of China (NSFC): 61876171 and Beijing Natural Science Foundation under Grant L182054.

## Footnotes

[1]The code and models are available on `https://github.com/blue-blue272/fewshot-CAN`

[2]We re-implement matching network with ResNet-12 as backbone on miniImageNet.

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
