[Supplementary Material]

# *Supplementary Material*
# Cross Attention Network for Few-shot Classification

**Ruibing Hou**[1,2]**, Hong Chang**[1,2]**, Bingpeng Ma**[2]**, Shiguang Shan**[1,2,3]**, Xilin Chen**[1,2]

[1]Key Laboratory of Intelligent Information Processing of Chinese Academy of Sciences (CAS),
Institute of Computing Technology, CAS, China
[2]University of Chinese Academy of Sciences, China
[3]CAS Center for Excellence in Brain Science and Intelligence Technology, China
ruibing.hou@vipl.ict.ac.cn, {changhong, sgshan,xlchen}@ict.ac.cn, bpma@ucas.ac.cn

This supplementary material includes more visualization results and comparisons between Cross Attention Network and the baseline model R12-proto-as (Section A), and an additional experiment on a dataset of cluttered scenes for few-shot classification (Section B).

## A    Network Visualization

For the qualitative analysis, we apply the class activation mapping [4] to different networks using images from tieredImageNet testing set. We compare the visualization results of CAN and the baseline (R12-proto-as) for the input image pairs from the same class. As shown in Fig. 1, we can clearly see that the class activation maps of our CAN can attend to the target object regions better than the baseline. As can be seen, the features of the baseline usually contain non-target objects, such as person, which leads to an inaccurate similarity estimation. On the contrary, with the proposed cross attention module, our CAN can learn to highlight the target object regions and aggregate features from them, producing a more accurate similarity measure.

Furthermore, we visualize the class activation maps for the input image pairs from different classes. As shown in Fig. 2, our CAN can still focus on the target object even if the input two images contain same common backgrounds, such as sky (Fig. 2 (a) and (b)) or lawn (Fig. 2 (c) and (d)). We argue that the meta-learner in the cross attention module can learn to ignore common backgrounds in the training phase, which avoids the interference of common backgrounds.

## B    Cluttered Scenes Evaluation

In this section, we evaluate our method on a more cluttered dataset, a scene recognition dataset miniPlaces365. We randomly select 100 classes with 600 images per class from the training set of Places365 to form miniPlace365. The classes are divided into 60 classes for training, 20 classes for validation and 20 classes for testing. The experimental setting and implementation details are the same with those of the miniImageNet dataset.

We compare our method to existing few-shot methods on miniPlace365 dataset. As shown in Tab. 1, CAN achieves significant gains, with an improvement up to $6\%$ on 1-shot setting and $7\%$ on 5-shot setting. The results demonstrate that our CAN is more efficient on cluttered scenes.

Table 1. Comparisons on 5-way classification on miniPlace365.

| Models | MN [3] | PN [1] | RN [2] | CAN |
|--------|--------|--------|--------|--------|
| 1-shot | 48.16 | 44.52 | 48.34 | **54.44** |
| 5-shot | 59.07 | 53.65 | 61.73 | **68.11** |

Figure 1. **Class activation mapping [4] visualization results.** We compare the visualization results of baseline with CAN (baseline with cross attention module). The input images of a pair are from the same testing class. As seen, our CAN is robust to the non-target objects. Warmer color with higher value.

Figure 2. **Class activation mapping [4] visualization results of CAN.** The input images of a pair are from different testing classes with same common backgrounds, *e.g.*, sky or lawn. As seen, our CAN is robust to the common background. Warmer color with higher value.