[Reviews · NeurIPS 2019]

Reviewer 1



Post rebuttal ========== I’d like to thank the authors for performing the additional ablation, comparisons via visualizations, and experiment in a cluttered environment, as I suggested in my reviews. I think these additional results would be good additions (to the Appendix at the very least) and strengthen the paper. I continue to recommend acceptance. I do agree with R3 though that the proposed transductive method is very similar to previous works for semi-supervised learning, and it would be useful to be more clear about this in the writing. Before rebuttal ============ Summary This paper introduces a state-of-the-art approach to few-shot classification. There are two orthogonal components proposed: the first influences the embedding function applied to the images of an episode, and the second introduces a strategy for using the query set of each episode in a transductive manner as additional unlabeled data for refining the within-episode classifier. The insight behind their embedding method is that it is useful to process each query example differently in light of the given support set, instead of processing each example of the episode independently. The motivation is that this may allow to identify objects that are common between a query example and the support set which are likely to be the target objects that the classification decision should be based on. Otherwise, it may be the case that the feature extraction emphasizes the identification of objects that are in the meta-training set, even if they are background (not target objects) for a particular episode. Within each episode, they first compute a prototype per class by averaging the (standard) embeddings of the support examples of each class. They also embed each query example independently to start with. Then, in the embedding space of the last convolutional layer, they compute for every prototype p and query example q, the channel-wise cosine similarity at each of the height x width spatial locations of that layer. This yields an m x m correlation matrix for the specific pair of class prototype and query example, where m = height x width. This correlation matrix is used for determining an attention map (over the spatial locations) for the processing of the prototype and one for the processing of the query example. These maps are obtained via a m x 1 convolution operation over the rows and columns of the correlation matrix, respectively. This convolutional layer is referred to as the meta-fusion layer and is meta-learned (i.e. updated based on the loss on the query set of each meta-training episode). Meta-learning this layer encourages the resulting attention mechanism to emphasize those features that are most discriminative for the classification task at hand. Once the attention maps are obtained, the prototype and the query example are re-embedded using the updated feature maps. The loss of the episode is then the nearest-neighbor based classification of the query examples. This episode-specific loss is combined with a global classification loss (over all classes of the training set) in a joint training fashion. Their proposed transductive strategy, in short, consists in treating the most confident (unlabeled) query examples of each episode as additional labeled examples that can be used to compute the class prototypes. The threshold that the confidence level of a query point must surpass in order to contribute to prototype creation becomes more permissive throughout training via a manually-defined schedule. This method achieves state-of-the-art on common datasets. Further, they run some ablations since their method involves a few components that are orthogonal to the proposed architecture and are not standard, namely joint training, and transductive treatment of the query set. For measuring the effect of the former, they experiment with a standard Prototypical Network and with a Prototypical Network that was jointly trained with the usual episodic query loss as well as a global classification via a softmax layer with as many outputs as total training classes (in the same way as they train their method). The fact that the latter outperformed the former significantly indicates that this process may be useful more generally for training meta-learning models. For assessing the benefit of their transductive approach, they applied it on other models too and showed that they also notably benefit from it, which is also an interesting result. Comments 1) I really like the type of exploration done in Figure 1 where the authors attempt to visualize which parts of the image their proposed method (and a baseline) pay attention to. I would argue that simply looking at classification accuracy results is not sufficient to conclude that the reason for improvement witnessed here is solely due to addressing the identified weakness (since unfortunately there are other confounding factors). Further, orthogonally to the accuracy results, it is an interesting finding if standard approaches indeed suffer from this and the proposed method provides a remedy. I would therefore focus on these qualitative results more, and explain in the main text (not just the appendix) exactly how those visualization are created, and show those results for various models. 2) Somewhat related to the previous point: Pure metric-based models like Prototypical Networks lack an explicit mechanism for adaptation to each task at hand and it therefore seems plausible that they indeed suffer from the identified issue. However, it is less clear whether (or to what extent) models that do perform task-specific adaptation run the same danger. Intuitively, it seems that task adaptation also constitutes a mechanism for modifying the embedding function so that it favours the identification of objects that are targets of the associated classification task. By task adaptation here I’m referring either to gradient-based adaptation (as in MAML and variants) or amortized conditioning-based adaptation (as in TADAM for example). Therefore, it would be very interesting to empirically compare the proposed method to these other ones not only in terms of classification accuracy but also qualitatively via visualizations as in Figure 1 that show the areas of the image that a model focuses more for making classification decisions. 3) Suggestion for the transductive framework: In Equation 8, it might be useful to incorporate the unlabeled examples in a weighted fashion instead of trusting that every example whose confidence surpasses a manually-set threshold can safely contribute to the prototype of the class that it is predicted to belong to. Specifically, the contribution of an unlabeled example to the updated class prototype can be weighted by the cosine similarity between that unlabeled example and that prototype (normalized across classes) and maybe additionally by the confidence c_b^q. This might slightly relieve the need to find the perfect threshold, since even if it is not conservative enough, a query example will be prohibited by modifying a prototype too much. An example of this is in Ren et al. [1] when computing refined prototypes by including unlabeled examples. 4) It seems that the weakness that this method is addressing would be more prominent in images comprised of multiple objects, or cluttered scenes. It would be very interesting to compare this approach to previous ones on few-shot classification on such a dataset! 5) For more easily assessing the degree of apples-to-applesness of the various comparisons in the tables, it would be useful to note which of the listed methods use data augmentations (as until recently this was not common practice for few-shot classification), what architecture they use, and what objective (most are episodic only but I think TADAM also performs joint training as the proposed method). 6) Another difference between the proposed approach and previous Prototypical Network-like methods is that the distance comparisons that inform the classification decisions are done in a feature-wise manner in this work. Specifically, when comparing embeddings a and b, for each spatial location, the distance between the feature vectors of a and b at that location is computed. The final estimate of the distance between a and b is obtained by aggregating those feature-wise distance estimates over all spatial locations. In contrast, usually the output of the last embedding layer is reshaped into a single vector (of shape channels x height x width) and distance comparisons of examples are made by directly comparing these vectors. It would therefore be useful to perform another ablation where a standard Prototypical Network is modified to perform the same type of distance comparison as their method. 7) Similarly to how the proposed transductive method was applied to other models, it would be nice to see results where the proposed joint training is also applied to other models, since this is orthogonal to the choice of the meta-learner too. References [1] Meta-Learning for Semi-Supervised Few-shot Classification. Ren et al. ICLR 2018.

Reviewer 2



This paper presents cross-attention for few-shot learning. It performs cross-correlation between the query image and each image in the support set. The obtained cross-attention maps are then used to gate the feature maps to obtain the final feature maps from both query and support images. I think the motivation is sound and proposed cross-attention maps for query and support images are novel. The experimental results validates the effectiveness of the proposed method. My major concern is on the time complexity of the proposed method, as it requires to conduct cross-correlation between query and every support images. In the comparision table, I think the authors should consider time complexity of different compared methods. I think the idea is novel and results show SOTA performance. My only concern is the computational complexity, as it needs to

Reviewer 3



Post Rebuttal: Thank you to the authors for their comments. I still believe that the discussion of the proposed transductive method should cite previous work in self-training methods for semi-supervised learning, as this is essentially what the authors have proposed for taking advantage of unlabeled query data. I am happy that the authors are committed to releasing the code for their method. Before Rebuttal: Summary This paper proposes a modification of embedding-based few-shot learning methods, where instead of embedding support and query set items independently, each prototype embedding is computed conditional on a query item and each query item is computed conditional on a prototype. Specifically, the conditional embedding takes the form of an attention module which is computed using a correlation layer and a fusion layer. The idea is that this type of attention allows the embedding to highlight specific objects with respect to another input. Classification is done via distance computed using this conditionally-computed embeddings and additionally a global classification layer, which classifies each query set item to a class label in the training set independent of the current episode. Lastly, a method for transductive inference is proposed, which involves taking the top-t classified query items and adding them to the support set with the predicted label and re-calculating the embeddings and repeating for a certain number of iterations. The method is evaluated on the mini-Imagenet and tiered-Imagenet benchmarks. Additional ablation studies are conducted to study the effects of different parts of the proposed model. Strengths 1. Paper is well-written and describes the proposed model well. 2. Experiments involve more than just pure results, as ablation studies validate different parts of model. 3. Achieves state-of-the-art results for Mini-Imagenet and Tiered-Imagenet (larger margin on Tiered-Imagenet) Weaknesses 1. Proposed model involves a lot of complicated moving parts - it is not clear whether it'll be easily reproducible given that it is so complicated. 2. I don't believe the proposed transductive method is very novel as I believe its related to a common way to incorporate unlabeled data in semi-supervised methods (see self-training methods in semi-supervised learning).

[Author Response · NeurIPS 2019]

We thank the reviewers for their positive comments on the novelty and performance improvement. We will release the
code for paper reproduction and facilitating using and building upon this method as **R1** and **R3** suggested.

**R1: *Qualitative comparisons via visualization to other meta-learners, especially to task-adaptive meta-learners.***

**A:** We compare the visualization results of CAN to other meta-
learners, Relation Network (RN)[33], MAML [6] and TADAM [23].
As shown in Fig. 1 (a), the features of RN usually contain non-target
objects since it lacks an explicit mechanism for feature adaptation.
MAML performs gradient-based adaptation, which makes the model
merely learn some conspicuous discriminative features in the sup-
port images without deeping into the intrinsic characteristic of the
target objects. As shown in Fig. 1 (b), MAML attends to *ship* for the
*groenendael* support image to better distinguish it from the *golden*
*retriever* category, resulting in a confusing location and misclassi-

Figure 1. Class activation mapping (*Cam*) visualization on a 5-way
1-shot task with 1 query sample per class.

fication of the *groenendael* category. TADAM performs task-dependent adaptation and applies the **same** adaptive
parameters to all query images of a task, thus it is difficult to locate different target objects for different categories. As
shown in Fig. 1 (c), TADAM mistakenly attends to the *dog* for *worm fence* query image. In contrast, CAN processes the
query samples with **different** adaptive parameters, which allows it to focus on the different target objects for different
categories shown in Fig. 1 (d). We will add these qualitative comparisons into the main text in the final version.

**R1: *Ablation that modifies a standard Prototypical Network to use the proposed feature-wise distance metric.***

**A:** Following your suggestion, we compare the standard Prototypical Network (PN) with Prototypical Network
using feature-wise distance metric (PN-F) on miniImageNet. PN-F only brings a marginal improvement while CAN
significantly outperforms it (PN/PN-F/CAN: 1-shot accuracy: 61.30/61.94/**63.95**, 5-shot accuracy: 76.70/76.91/**79.44**).
The results further verify that the significant improvement of CAN to PN is due to the proposed cross attention module.

**R1: *Apply the proposed joint training schema to other commonly-used meta-learners.***

**A:** We try another two meta-learners, Matching Network (MN) [36] and Relation Network (RN) [33], to further verify the
effectiveness of the proposed joint learning schema . We re-implement MN and RN with

Table 1. Performance with joint training.

| models | MN | MN-JT | RN | RN-JT |
|---|---|---|---|---|
| 1-shot | 55.29 | **59.14** | 51.25 | **54.29** |
| 5-shot | 67.74 | **73.81** | 64.45 | **67.58** |

ResNet12 as backbone on miniImageNet. As shown in Tab. 1, our joint training schema
(-JT) significantly improves the performance with respect to different meta-learners.

**R1: *Experiment on a dataset of cluttered scenes for few-shot classification.***

**A:** Following your suggestion, we use a more cluttered dataset, a scene recognition dataset miniPlaces365 [1]. A scene
image usually contains multiple objects, while not all the objects are
related to this scene. Therefore, it requires the models to accurately
locate the target objects for correct classification. We compare CAN
to MN, RN and PN with the same backbone and joint-training schema
on miniPlace365. CAN achieves more gain, with an improvement up

Figure 2. *Cam* visualization on input pairs from same class.

to $6\%$ (MN/RN/PN/CAN: 1-shot: 48.16/44.52/48.34/**54.44**). The results demonstrate that CAN is more efficient on
cluttered scenes. For qualitative analysis, we compare the visualization results in Fig 2. As can be seen, other methods
usually highlight non-target objects, while CAN can attend to the targets among multiple objects of the input images.

**R2: *Compare time complexity to other methods.***

**A:** (***i***) Our cross attention module (CAM) only increases marginal time cost. The cross-correlation maps between a
query image and all support images can be simply worked out by one matrix multiplication, which is lightweight when
it is used in high-level, sub-sampled feature maps. To illustrate the extra cost of CAM, we compare the time cost of the
backbone for feature extraction and CAM for cross-correlation estimation in a 5-way 1-shot task. The backbone takes
0.041s for a query data, while CAM only takes 0.002s, equivalent to only $\sim 4\%$ relative time increase over the backbone.
(***ii***) Tab. 2 further compares the time cost of our method to others. Some methods [36,31,33,13,6] use a 4-layer *Conv* as
the backbone thus take relatively lower time cost. Even though, our CAN is still comparable even superior to these
methods in term of time cost, with a performance improvement up to $10\%$. The others use the same backbone as CAN,
but require following up modules such as model update per task [32,12], gradient-based parameter generation [19], or
expensive condition generation [23], which all incur more time overhead than CAM. Overall, Tab. 2 shows that CAN
outperforms other methods without excessive overhead. We will report the time complexity of different methods in the
comparison table (Tab1 in the main paper) in the final version.

Table 2. Time overhead of different methods. All times are reported per query data in a 5-way 1-shot task on one NVIDIA 1080Ti GPU.

| model | MN[36] | PN[31] | RN[33] | DN4[13] | MAML[6] | MTC[32] | MetaOptNet[12] | adaNet[19] | TADAM[23] | CAN |
|---|---|---|---|---|---|---|---|---|---|---|
| test time (s) | 0.021 | 0.018 | 0.033 | 0.049 | 0.103 | 2.02 | 0.096 | 1.371 | 0.079 | 0.044 |

52
51
**R3: *Novelty of transductive method and Release the code for reproduction.***
**A:** Thank you for your positive comments on the writing and performance improvement. (***i***) For the second contribution,
the transductive method, we are the first to explore the idea that incorporates the unlabeled **query data** to **refine**
**prototypes** in a meta-learning setup. We demonstrate it is effective to alleviat the *low-data* problem on transductive
few-shot setting, which outperforms prior work [15] by a large margin, up to $8\%$ improvement. (***ii***) All implementation
details of CAN are given in the 'Experiment Setup' section. We will release the code and trained models for paper
reproduction. In addition, we will submit the code with the camera-ready version once this paper is accepted.

are divided into 60 classes for training, 20 classes for validation and 20 classes for testing. The input images size is $84 \times 84$.

## Footnotes

[1] We randomly select 100 classes with 600 images per class from the training set of Places365 to form miniPlace365. The classes


[Meta-Review · NeurIPS 2019]

This paper was reviewed by three experts in the field and received 677 recommendations. Based on the reviewers' feedback, the decision is to recommend the paper for acceptance to NeurIPS 2019. The reviewers did raise some valuable concerns especially on the citing related works for the transductive part and the additional experiments and analyses suggested (and provided in the rebuttal). These questions should be addressed in the final camera-ready version of the paper. The authors are encouraged to make the necessary changes to the best of their ability. We congratulate the authors on the acceptance of their paper!